# HIRM v1.0: A hybrid impulse response model for climate modeling and uncertainty analyses

Kalyn Dorheim [1], Steven Smith [1], Ben Bond-Lamberty[1]

[1] Joint Global Change Research Institute, Pacific Northwest National Laboratory, College Park, MD 20740, United States of America

*Correspondence to*: Kalyn Dorheim (kalyn.dorheim@pnnl.gov)

**Abstract.** Simple climate models (SCMs) are frequently used in research and decision-making communities because of their flexibility, tractability, and low computational cost. SCMs can be idealized, flexibly representing major climate dynamics as impulse response functions, or process-based, using explicit equations to model possibly nonlinear climate and Earth system dynamics. Each of these approaches has strengths and limitations. Here we present and test a hybrid impulse response modeling framework (HIRM) that combines the strengths of process-based SCMs in an idealized impulse response model, with HIRM's input derived from the output of a process-based model. This structure enables the model to capture some of the major nonlinear dynamics that occur in complex climate models as greenhouse gas emissions transform to atmospheric concentration to radiative forcing to climate change. As a test, the HIRM framework was configured to emulate the total temperature of the simple climate model Hector 2.0 under the four Representative Concentration Pathways and the temperature response of an abrupt four times $CO_2$ concentration step. HIRM was able to reproduce near-term and long-term Hector global temperature with a high degree of fidelity. Additionally, we conducted two case studies to demonstrate potential applications for this hybrid model: examining the effect of aerosol forcing uncertainty on global temperature, and incorporating more process-based representations of black carbon into a SCM. The open-source HIRM framework has a range of applications including complex climate model emulation, uncertainty analyses of radiative forcing, attribution studies, and climate model development.

## 1 Introduction

Climate models encompass a diverse collection of approaches to representing Earth system processes at various levels of complexity and resolution. The most complex are the Earth System Models (ESMs): highly detailed representations of the physical, chemical, and biological processes governing the Earth system at high spatial and temporal resolution (Hurrell et al. 2013). These models are computationally expensive and therefore can only be run for a limited number of scenarios. Slightly less complex and more computationally efficient are the Earth System Models of Intermediate Complexity (EMICS) (Stocker, 2011). Finally, Simplified Climate Models (SCMs) sacrifice process realism but are computationally inexpensive (van Vuuren et al., 2011). Although SCMs are generally low resolution in space and time, they have a wide range of applications including emulation (Dorheim et al. 2019); probabilistic estimates demanding thousands of separate model runs (Stainforth et al. 2005;

Webster 2012); factor separation analysis (Mheel et al., 2007); and Earth system model development and diagnosis (Meinshausen et al., 2011).

SCMs vary in complexity. Process based SCMs such as Hector (Hartin et al. 2015) and MAGICC (Meinshausen et al., 2011) consist of systems of equations that represent, albeit in highly simplified form, carbon cycle and climate dynamics. Other
SCMs are more abstract, consisting of a few highly parameterized equations. Some of the more idealized SCMs (*sensu* Millar et al. 2017) use impulse response functions (IRFs) to approximate climate dynamics (Millar et al., 2017). IRF-based SCMs are themselves diverse; some are highly idealized, such as the Impulse Response Function used in the Fifth IPCC Assessment Report (Myhre et al., 2013) (AR5_IR), while others are quasi process-based, only using IRFs to approximate linear climate dynamics, with the rest of the climate system represented by process-based equations (Strassmann and Joos 2018; Smith et al.
2018; and Joos and Bruno 1996).

One of the fundamental differences between process-based SCMs and idealized IRF-based SCMs is in their representation of the important nonlinear climate dynamics occurring during the evolution of emissions to climate impacts. Process-based models (whether SCMs or ESMs) have equations that represent emissions accumulating as concentrations, which in turn affect
the energy (radiative forcing) resulting in climate changes (most prominently, temperature change) (Harvey et al. 1995; Claussen et al. 2002). The system of equations used by process-based SCMs represents some, though not all, of the more complex and often nonlinear dynamics observed in the Earth system. These dynamics include interactions between atmospheric chemical constituents (Wigley et al. 2002); non-linear relationships between greenhouse gas concentrations and energy absorption, i.e. radiative forcing (Shine et al. 1990 and Myhre et al. 1998); and carbon-climate feedbacks such as ocean
surface $CO_2$ uptake (Wenzel et al. 2014, Tang Riley 2015, and Heinmann and Reichstein 2008). Comprehensive process-based SCMs such as Hector and MAGICC have thousands of lines of code and take significant effort to expand. On the other extreme, simple impulse response models can be expressed in a few equations and are readily implemented, but these simplifications can produce biases in results (van Vuuren et al. 2011, Schwarber et al. 2019). We discuss here a framework that can be used as a testbed for SCM development and analysis.


In this manuscript we document and demonstrate a highly idealized IRF-based framework. This modeling framework is configured using output from a process-based model to capture nonlinear and complex climate dynamics, we refer to it as a hybrid impulse response modeling (HIRM) framework. HIRM was configured using the open source, object-oriented, process-based SCM Hector v 2.3.0, although in theory it could potentially use information from any climate model (ESM, EMIC,
SCM). The first two experiments in this paper demonstrate HIRM's ability to accurately reproduce global mean temperature, including the temperature response to large climate system perturbations. We also demonstrate the potential utility of this framework in an uncertainty analysis and examine how changing the response function for black carbon impacts HIRM output. We discuss the implications of these results as well as potential future uses of this framework.

## 2 Methods

### 2.1 Parent Model Description

In this study we used Hector v 2.3.0 as the parent model, providing both of HIRM's primary and only inputs. We selected Hector because it is open source, well documented, fast-executing, and has a structure that makes it easy to obtain 'clean' IRFs from model runs (Schwarber et al. 2019). (As noted above, however, HIRM can be coupled with any parent model that can provide its inputs.) Hector has been well documented (Hartin et al. 2015), but we provide a brief summary here.

Hector (Hartin et al. 2015) is an open source, process-based SCM carbon-climate model available at https://github.com/jgcri/hector. The model is written in C++ and has an object-oriented structure, allowing for substitutions of different model components; it has both internal and external automated testing, e.g. enforced unit-checking, that provides robustness and quality assurance. Hector models carbon and energy flows between the ocean, atmosphere, and terrestrial biosphere, starting with a preindustrial steady-state system that is then perturbed by anthropogenic emissions provided as input files. The model runs on an annual timestamp, although the carbon cycle as an adaptive-timestep solver to ensure smooth numerical changes when fluxes (primarily ocean uptake) are large. The terrestrial carbon cycle is divided into biota, litter, and soil across multiple biomes; the ocean features surface, intermediate, and deep pools in different hemispheres, with heat uptake governed by an implementation of the DOECLIM (Kriegler 2005; Urban et al. 2014) 1-dimensional heat diffusion sub-model. Hector models the dynamics of 37 different radiative forcing agents. The total radiative forcing in turn affects global temperature change, with all of Hector's radiative forcing agents exhibiting the same temperature response to change in radiative forcing. In effect, Hector can be considered to interpret forcing assumptions as Effective Radiative Forcing values, which are more closely related to surface temperature changes than the previously used values for stratospheric-adjusted radiative forcing (Richardson et al. 2019). This has no impact on the model dynamics that are our focus here, and only impact how numerical values are selected as input settings. Note that Hector also assumes that the temporal shape of the response function is the same for all forcers, a simplifying assumption that has consequences for HIRM configuration, but also the consequences of which we examine below.

### 2.2 HIRM Description

HIRM's total atmospheric temperature response is calculated as the sum of the Green's function of a temperature response to a radiative forcing perturbation with radiative forcing time series, an approach taken by many SCMs (Joos et al. 1999; Van Vuuren 2011; Millar et al. 2015; and Boas 2006). By relying on a process-based climate model to compute RF values, HIRM is able to use a linear IRF in a simple impulse response model and capture the major nonlinear dynamics between the emissions to radiative forcing calculations by using radiative forcing time series as input data.

HIRM calculates the atmospheric temperature change from preindustrial temperature ($T$) as the sum of the temperature contribution from individual forcing agents $T_i$ Eq. (1):

$$T(t) = \sum_{i=1}^{n} T_i(t), \tag{1}$$

Here the individual temperature contribution is equal to the convolution of the radiative forcing time series $RF_i$ with the temperature response to a radiative forcing pulse $IRF_i$ for a single radiative forcing agent Eq. (2).

$$T_i(t) = \int_{t_0}^{t} RF_i(t')IRF_i(t - t')dt', \tag{2}$$

The method we used to obtain $RF_i$ and $IRF_i$ for HRIM relies on output from a parent process-based model. The subsequent sections discuss how we obtained $RF_i$ and $IRF_i$ specifically from Hector. It is important to note that while HIRM can be set up with unique IRFs for each radiative forcing agent (as demonstrated below), this was not done in this application since Hector uses one IRF for all species.

HIRM is an open source R package (https://github.com/jgcri/hirm) with Doxygen-style comments, unit tests, and online documentation via pkgdown (Wickham and Hesselberth 2020). The online documentation available at https://jgcri.github.io/HIRM/ documents all of the package functions and links with a vignette(example) that demonstrates how to set up and run HIRM. The package contains all of the IRFs and RF inputs used in this manuscript that can be used in a customizable configuration matrix to set up and run HIRM.

**2.3 IRF Derivation**

As previously mentioned, one of Hector's assumptions is that all of Hector's radiative forcing agents elicit the same temperature response to a change in radiative forcing. Even though HIRM can use a unique IRF for each radiative forcing agent, for the purposes of HIRM validation exercises in this study, HIRM's setup must be analogous to that of its parent model Hector. In this study we configured HIRM with a single IRF that characterizes Hector's temperature response to all of its 37
radiative forcing agents, derived from a reference run and a black carbon (BC) emissions perturbation run of Hector. In Hector, BC emissions are converted directly to radiative forcing, and therefore an emissions pulse of BC is analogous to a radiative forcing pulse. BC was chosen as the forcing agent since there are no gas-cycle or forcing interactions with other species within Hector, making it straightforward to derive the IRF, but other forcing agents could have been selected for the perturbation run. During the reference model run Hector was driven with the RCP 4.5 scenario, while for the perturbation model run BC
emissions were doubled relative to RCP 4.5 BC emissions in a single year. RCP 4.5 $CO_2$ concentrations were prescribed during these runs, suppressing Hector's normal carbon cycle temperature feedbacks.

For the two validation experiments we did not include carbon-cycle climate feedbacks into the IRF as we wanted to examine the relative importance of non-linearities in emission to forcing calculations, at least as represented in Hector, as compared to
non-linearities in Hector's forcing to temperature calculations (as represented within DOECLIM). For this reason the IRF should represent only the response of temperature to radiative forcing; otherwise, the temperature response from these feedback mechanisms would be incorporated into the IRF, which would then be doubled-counted as forcing time series are being used as inputs. For the replication experiments we also focus on reproducing Hector temperature without carbon-climate feedbacks. Other applications of HIRM may require IRFs that include the temperature response from carbon-cycle feedbacks.

The temperature response ($T_{response}$) to the BC emissions perturbation is equal to the difference between the reference ($T_{ref}$) and perturbation temperature $\left(T_p\right)$ Eq. (3), with the perturbation occurring at year $t_0$:

$$T_{response}(t - t_0) = T_p(t - t_0) - T_{ref}(t - t_0), \tag{3}$$


The temperature response to a radiative forcing perturbation was calculated by dividing the temperature response to the emissions perturbation by the size of the radiative forcing pulse Eq. (4). The size of the radiative forcing pulse ($X_y$) was set equal to the difference in radiative forcing between the reference and emissions perturbation runs (described in the paragraph above) in the perturbation year:


$$IRF_i(t - t_0) = T_{response}(t - t_0)/X_y, \tag{4}$$

This IRF had a length of 300 years, in order to ensure the IRF was long enough to be convolved with the RF inputs; the end of the IRF was extrapolated with an exponential decay function to a length of 3000 years with a decay constant of 0.20.
Extending the length of the IRF prevents the IRF from being padded with zeros and having to truncate the RF inputs.

The majority of Hector's temperature response to a radiative forcing pulse occurs within the first 50 years after the perturbation (Fig. 1). The strongest response occurs during the perturbation year itself, with a maximum value of 0.09 ($°C\ W^{-1}m^{-2}$); by year 35 the temperature response has decreased by 97% and continues to approach zero for the remainder of the IRF. This IRF is
used in both of the validation experiments and case studies except where noted.

## 3 Validation Experiments

### 3.1 Replication of RCP Results

Emulation is used to validate HIRM by illustrating that the HIRM framework reproduces the dynamics of a process-based SCM with a minimal loss of information. If HIRM can accurately reproduce or emulate the atmospheric temperature of a more complex, process-based-model such as Hector, then we assume that HIRM is able to capture important non-linear dynamics of the climate system using this setup, at least to the extent these are captured in the SCM. Conversely, if HIRM is unable to reproduce Hector's global temperature outputs, this would indicate that important processes are not being captured by the HIRM framework.

In the first validation experiment, HIRM was set up to reproduce Hector temperature for RCP 2.6, RCP 4.5, RCP 6.0, and RCP 8.5. HIRM was configured for each RCP scenario with a single IRF derived from Hector (fig. 1) together with a complete set of time series from Hector's 37 radiative forcing agents. The radiative forcing time series for these validation experiments came from Hector output from RCP 2.6, 4.5, 6.0, and 8.5 with prescribed $CO_2$ concentrations. The global mean temperature outputs from Hector driven with RCP 2.6, RCP 4.5, RCP 6.0, and RCP 8.5 were saved and used as validation data for HIRM.

HIRM was able to emulate Hector's temperature for the four RCPs with a minimal loss of information (Fig. 2a). The difference between HIRM and Hector total temperature, measured as the root mean squared error (RMSE), was $1.3 \times 10^{-9}$ °C (Fig. 2a) for each RCP scenario. The cumulative percentage difference between HIRM and Hector temperature was 0 % (rounded from $1.0 \times 10^{-5}$; other 0% results are similar) for each RCP scenario.

### 3.2 Replication of 4X $CO_2$ Results

The second validation experiment tested HIRM's ability to reproduce Hector's temperature response to an abrupt four times $CO_2$ concentration step. The abrupt four times $CO_2$ concentration step is a test commonly used by climate modelers to understand the climate system's response to $CO_2$ (Taylor et al 2012). In this experiment HIRM was set up with the Hector derived IRF and a RF input from an abrupt four times $CO_2$ concentration step. The radiative forcing time series was obtained from Hector runs following the CMIP5 protocol (Taylor et al. 2012). HIRM's radiative forcing time series input was the difference in Hector radiative forcing from Hector driven with a constant $CO_2$ concentration of 278 ppm and Hector driven with a $CO_2$ concentration of 278 ppm until year 2010 when the $CO_2$ concentration increased by a magnitude of four and remained constant for the rest of the run. The difference in Hector's global mean temperature anomaly between the constant reference run and the perturbed step run was then compared with HIRM's output.

HIRM reproduced Hector's abrupt four times $CO_2$ concentration step temperature response with a high degree of accuracy (Fig. 2b). The RMSE between HIRM and Hector temperature output from the abrupt $CO_2$ concentration step was $1.5 \times 10^{-19}$ °C with a cumulative percent difference of 0%. The abrupt $CO_2$ concentration step is a standard diagnostic test used to examine

climate model responses (Taylor et al. 2012; Eyring et al. 2016). Since HIRM was able to accurately emulate Hector's temperature response to a large step perturbation we conclude that the majority of the nonlinearities within Hector are occurring during the emissions-to-radiative forcing portion of the emissions-to-temperature causal chain. While this is to be expected from the general principles of SCMs, it nonetheless provides a useful check that our understanding of the parent model's behavior is correct.

## 4 HIRM Application Case Studies

### 4.1 Aerosol Uncertainty Case Study

Uncertainties in the magnitude of historical and future radiative forcing effects continue to be a crucial challenge for climate science research, and this is particularly true for aerosol effects (Forest 2018). In this first case study HIRM was used to explore a range of future temperature change when accounting for uncertainty in some aerosol radiative forcing effects, specifically black carbon (BC), organic carbon (OC), indirect $SO_2$ effects ($SO_2i$), and direct $SO_2$ effects ($SO_2d$). To do so, HIRM was again set up to recreate Hector's RCP 4.5 temperature. In this analysis, BC, OC, $SO_2i$, $SO_2d$ RF inputs were varied. (Aerosol cloud indirect effects are represented in Hector as a function of $SO_2$ emissions only, so we refer to that as $SO_2$ indirect forcing.) We present a simple demonstration of the model in this case study and note that we have not produced probabilistic results but an illustrative range of temperature pathways that result from aerosol uncertainties (e.g. Smith and Bond 2014). A full probabilistic analysis would also involve varying additional parameters, such as climate sensitivity, ocean heat update, and carbon-cycle parameters.

The aerosol uncertainty scalers were generated from the 2011 aerosol radiative forcing ranges reported in IPCC AR5 8.SM table 5 (Myhre et al. 2013). The BC, OC, $SO_2i$, and $SO_2d$ radiative forcing IPCC ranges were individually sampled at intervals of 0.04 W m$^{-2}$ in 2011 (Table 1), resulting in a total of 29000 times uncertainty scalar combinations. Default HIRM 2011 BC, OC, $SO_2i$, and $SO_2d$ radiative forcing values were then divided by the values sampled from the respective IPCC ranges to obtain the uncertainty scalers.

HIRM was set up to run every possible combination of the scaled RF time series, a total of 29000 times. This created an ensemble of uncertainty runs, whose results were constrained (i.e., filtered) by historical radiative forcing and temperature. HIRM total radiative forcing was constrained to match IPCC historical estimates in radiative forcing and temperature change. The 2011 aerosol (SO2i, SO2d, BC, + OC) radiative forcing was constrained to pass through an uncertainty range [-1.66 to 0.14 Wm-2] (similar to Myhre et al. 2013, but adjusted to account for nitrate and dust forcing and empirical constraints, see discussion in Smith and Bond 2014). HIRM temperature trend was calculated as the slope of a linear regression and then

compared to the observed temperature trend range of [0.65 to 1.1] °C over 1880–2012 reported by Hartmann et al. 2013. Cases that did not meet these constraints were removed (see Fig. 3).

We found that the historical constraints had an unequal impact on the scaled radiative forcing impacts. The temperature at the end of the century for the unconstrained ensemble ranged over 2.5°C – 3.1°C; incorporating the historical constraints into the uncertainty analysis narrowed uncertainty in future temperature to 2.7°C – 2.9°C (Fig. 3). The historical constraints had different impacts on the sampled aerosol uncertainty scalers. All of the sampled OC scalers passed through the historical constraints (Fig. 4b), while the constraints had a modest effect on the OC, BC, and $SO_2d$ scalers (Fig. 4a, b, and c).

The historical constraints have the most noticeable effect on the SO2i uncertainty scalers. This is because of the large absolute magnitude of the uncertainty in aerosol indirect effects (Myhre et al., 2013), which results in a large role for assumptions about the strength of aerosol indirect cooling (Tomassini et al 2007, Meinshausen et al. 2009). This shows that strong (negative) aerosol indirect forcing is consistent with only a few numerical combinations of forcing values from other species, at least for default Hector climate system parameters. The sample analysis using HIRM illustrates how this modeling framework can be utilized to calculate the range of past and future temperature changes under assumed uncertainty in aerosol radiative forcing.

**4.2 HIRM as a Tool for Development Case Study**

Radiative forcing effects from aerosols are complex (Fan et al. 2016, Bond et al. 2013), and while the physics driving these complexities have been incorporated into ESMs, they are not considered in most SCMs. For example, consider black carbon (BC): unlike cooling effects from aerosols that scatter shortwave radiation back into space, BC heats within the atmosphere, and also at the surface when deposited on snow or ice, potentially contributing to both cloud indirect cooling and heating effects (Bond et al. 2013). It can also increase cloud amounts, as BC atmospheric heating stabilizes the atmospheric thermal profile (Bond et al. 2013). Experiments conducted with ESMs have found large differences in the response to a step change in BC emissions compared to a step change in $CO_2$ (Sand et al. 2015; Yang et al. 2019).

Incorporating these dynamics into Hector would be a nontrivial task, but HIRM can be used to estimate what effect they would have on the model's global temperature. For this case study, HIRM was set up to emulate Hector RCP 4.5 as before, but with one difference: instead of pairing the BC RCP 4.5 RF time series with Hector's single IRF, the BC RCP 4.5 RF time series was paired with a BC-specific IRF. Since HIRM is set up with a BC-specific IRF, the results will no longer be equivalent to Hector's. Instead, the results illustrate what Hector's temperature could be if the BC dynamics were modified.

The BC-specific IRF was derived using output from a study that performed BC emission step tests with the ESM NorESM-1 (Sand et al. 2015). Mathematically, the derivative of a step response is equal to the impulse response function, and therefore we can derive an impulse response function from the step response results reported in the Sand et al. ESM experiment. The

temperature response to a BC step in ESM experiments is well fit by a single exponential approach to a constant response (see Yang et al. 2019 for details). We fit the Sand et al. (2015) abrupt BC step response as:

$$T(t) = A\,(1 - e^{\frac{-t}{\tau}}),\tag{5}$$

The results of a nonlinear optimization of this function returned values of and $\tau$ that were 1.8 °C and 2.1 years, respectively. These optimal values were used in Eq. (6), the differentiated form of Eq. (5), to provide a numerical BC temperature impulse response function corresponding to the Sand et al. (2015) result:


$$R_t(t) = \frac{A}{\tau}\,e^{\frac{-y}{t}}dt,\tag{6}$$

The numerical result of Eq. (6) is converted to a BC impulse response per unit forcing by dividing by the forcing from a 133 Tg BC emissions change (used in Sand et al. 2015) using Hector's default forcing per unit BC emission assumptions. With

this transformation we have replaced Hectors' default BC representation in HIRM with the Sand et al. temperature response in both magnitude and temporal behavior.

We found that the BC Sand et al. IRF has a weaker temperature response in the perturbation year and a more rapid decline in temperature response compared to Hector's global IRF (Fig. 5a). The maximum IRF response for the BC Sand et al. IRF is

0.06 (°C W$^{-1}$m$^{-2}$) which is 0.03 (°C W$^{-1}$m$^{-2}$) cooler than Hector's IRF. In addition, the BC. Sand et al. IRF approaches 0 (°C W$^{-1}$m$^{-2}$) faster than Hector's IRF. These differences are expected since the BC Sand et al. IRF was derived from the NorESM-1 ESM, meaning that this IRF incorporates the complex cooling and warming effects of BC emissions, the net warming over land as compared to no net warming over oceans (Sand et al. 2015). When HIRM was configured with the BC Sand et al. (2015) IRF the global temperature was lower by 0.2 °C from 1750 to 2100 under the RCP 4.5 scenario (Fig. 5b). Based on

these results, if Hector were modified to emulate this BC response, we predict that the model's global temperature would be cooler by approximately 0.2 °C in 2100.

We note the idea of different forcing agents has been around for quite some time. For example, this has been incorporated mechanistically for aerosols in the MAGICC model for around 30 years now (Wigley and Raper 1992), and more recently

inferred by Shindell (2014) from GCM results. Richardson et al. (2019) used separate response functions for CO2, CH4, solar insolation, and aerosols, although the differences in these response functions were not discussed. As further information on species-specific IRFs become available it will be important to quantify the consequences of these different IRFs using tools such as HIRM.

## 5 Discussion and Conclusion

In this paper we document and test HIRM, a framework that leverages the nonlinear dynamics of process-based SCMs within a computationally efficient, highly idealized linear impulse response model. Our two case studies demonstrate that HIRM can be used as a testbed to quickly examine the consequences of different model assumptions, and to estimate changes in parent model behavior from including new mechanisms. While other IRF-based models have incorporated nonlinear dynamics using a number of approaches (Hooß et al. 2001, Millar et al. 2017, ADD), HIRM is able to demonstrate nonlinear dynamics through

its use of exogenous forcing inputs from Hector. HIRM is available as an open source R package (available at https://github.com/JGCRI/HIRM), its computational flexibility and short run time make it particularly appropriate for uncertainty analyses and experimental SCM design.

We demonstrated that HIRM can be used to examine uncertainty within the climate system, and that incorporating a more

realistic BC temperature response into Hector has a significant impact on Hector's global temperature. If more studies corroborate the findings of Sand et al. (2015) and Yang et al. (2019) by observing shorter timescale responses for BC temperature dynamics across a number of ESMs and Atmosphere-Ocean General Circulation Models (AOGCMs), then SCM modeling groups will need to consider incorporating the BC temperature response dynamics into SCMs. Some SCMs, such as MAGICC 5.3 and MAGICC 6 (Wigely et al. 2002), already exhibit multiple temperate responses; interestingly, MAGICC has

a shorter timescale for the temperature response for aerosols (Schwarber et al. 2019), but the resulting response in MAGICC still has a longer timescale than that from the AOGCMs (Sand et al. 2015, Yang et al. 2019).

During the HIRM validation experiments we demonstrate that most of nonlinearities are in the emissions to forcing steps, in which the SCM calculates concentrations from emissions and radiative forcings from concentrations, relationships that widely

used (Etminan et al. 2016). In comparison the non-linearites in going from forcing to global-mean temperature are relatively minor in contrast. This implies that efforts to improve the representation of nonlinear behavior in SCMs should be focused on emissions-to-forcing processes. We note that we draw this conclusion by calibrating HIRM to a single process-based SCM; this finding should be verified using other models, including Earth System Models of Intermediate Complexity (Claussen et al. 2002). Such EMICs have more physically-based parameterizations but low levels of internal model noise, which would be

valuable for exploring the magnitude and nature of non-linearites in going from forcing to temperature. If this finding holds for a wider class of models, this would mean that a wide range of model responses to forcing could be quickly simulated using IRFs. Good et al. 2013 showed that SCMs based on step responses work fairly well for more reproducing General Circulation Model (GCM) results suggesting that the assumptions underlying HIRM are valid.

The case studies showcase HIRM's flexibility, which is based on HIRM's dependence on a parent model. Arguably this can be viewed as a limitation or a tradeoff allows HIRM to be used as a tool for rapid exploration. One limitation of this framework

is that interactions between forcing agents are not directly considered. For example, multiple species of aerosols may contribute to cloud indirect cooling effects. These interactions, however, are not well constrained (Fan et al. 2016) and, for many purposes where SCMs might be applicable, it is most important to be able to represent the overall (large) uncertainty range, rather than interactions among species that have yet to be definitively quantified. An effort to represent aerosol indirect effects semi-analytically (Ghan et al. 2013) demonstrated not only the multiple processes that are relevant but also the difficulty in understanding the drivers of the different forcing estimates from more complex models.

Insights gained from HIRM could be useful in future work applying impulse response functions in general and the design of simple climate models in particular. We suggest that improvements to simple climate models should focus on improving the representation of emission to concentration and concentration to forcing relationships. As we note above, however, it would be useful to also design comparisons with more complex models, perhaps EMICS given their lower noise and computational requirements, to determine the extent to which the temperature response to forcing in more complex models can be accurately represented by impulse response functions, particularly on 20-30 year time scales where GCM outputs are particularly noisy.

HIRM could also be used with data generated by other SCMs. This could be a useful way of decomposing differences in responses between SCMs (e.g. Nicholls et al. 2020) into differences in the emissions to forcing step compared to differences in the model's response to a forcing impulse. Similarly, HIRM could be used to examine the uncertainty due to the different forcing to temperature responses amongst SCMs (see Schwarber et al 2019 for examples of different forcing to temperature IRFs).

HIRM can be used as a testbed for future SCM development. As demonstrated here, the incorporation of a GCM-derived temperature response function for black carbon emissions results in a significantly different global mean temperature response (Figure 5). Exploration of the potential impact of such changes can be done quickly in HIRM to decide if changes should be incorporated into, for example, Hector. Incorporating such a change into the Hector model itself would be a more time and labor intensive process for several reasons. First, to incorporate this change into Hector one would need to decide how to physically interpret the faster BC response time seen in GCMs since Hector does not use impulse response functions directly. There is some debate if this is due to different response over land vs ocean, or if this is more closely related to differing hemispheric responses (Meinshausen et al. 2011, Shindell 2014, Sand et al. 2015). Further, explorations or model extensions using HIRM can be accomplished without a user having to understand Hector's code, dependencies, and coding standards.

Finally, this framework could also be used for analysis that requires capabilities not present in SCMs–for example, regional analysis. Regional temperature trends could be simulated by HIRM by incorporating the ratio of regional to global temperature responses for each forcing agent into HIRM (Sand et al. 2019 and Shindell and Faluvegi et al. 2009). This could be particularly valuable for a region such as the Arctic, where a variety of forcing agents, from regional sulfate (Acosta Navarro et al. 2016), local black carbon (Sand et al. 2013 and Yang et al. 2019), and global forcing changes, e.g. Arctic amplification, all may play

a role. This type of analysis could readily be accomplished using HIRM, including the wide range of uncertainty space that should be examined (e.g. Figure 3). Future research with HIRM could test IRFs set up with different climate sensitivity values and inputs from other process-based models.


### Code availability

The HIRM R package is available at https://github.com/JGCRI/HIRM with an online manual available at https://jgcri.github.io/HIRM/ . The package is also archived on Zenodo (https://doi.org/10.5281/zenodo.3756122). Code and results related to the discussion and conclusions of this paper are available on the Open Science Framework (OSF) at

https://osf.io/kmrj8/.

### Author contributions

S. Smith conceptualized the Hybrid Impulse Response Model (HIRM). K. Dorheim and B. P. Bond-Lamberty developed the project software. K. Dorheim wrote the manuscript with contributions from all co-authors.

### Competing interests

The authors declare that they have no conflict of interest.

### Acknowledgements

We thank Robert Link for invaluable insight regarding HIRM development as an R package and M. Sand for numerical model results. This research was supported by the United States Environmental Protection Agency.

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

| RF agent | Min. 2011 RF | Max. 2011 RF | Hector Default 2011 RF |
|----------|--------------|--------------|------------------------|
| BC | 0.05 | 0.87 | 0.40 |
| OC | -0.21 | -0.04 | -0.17 |
| $SO_2i$ | -1.2 | 0 | -0.60 |
| $SO_2d$ | -0.6 | -0.2 | -0.35 |

**Table 1: The minimum and maximum 2011 radiative forcing values from IPCC AR5 8.SM table 5 (Myhre et al. 2013). These values were used to obtain the min and max aerosol uncertainty scalers for four RF agents (BC, OC, $SO_2i$, and $SO_2d$). Along with the 2011**
**RF of the default configuration of HIRM/Hector for RCP 4.5.**

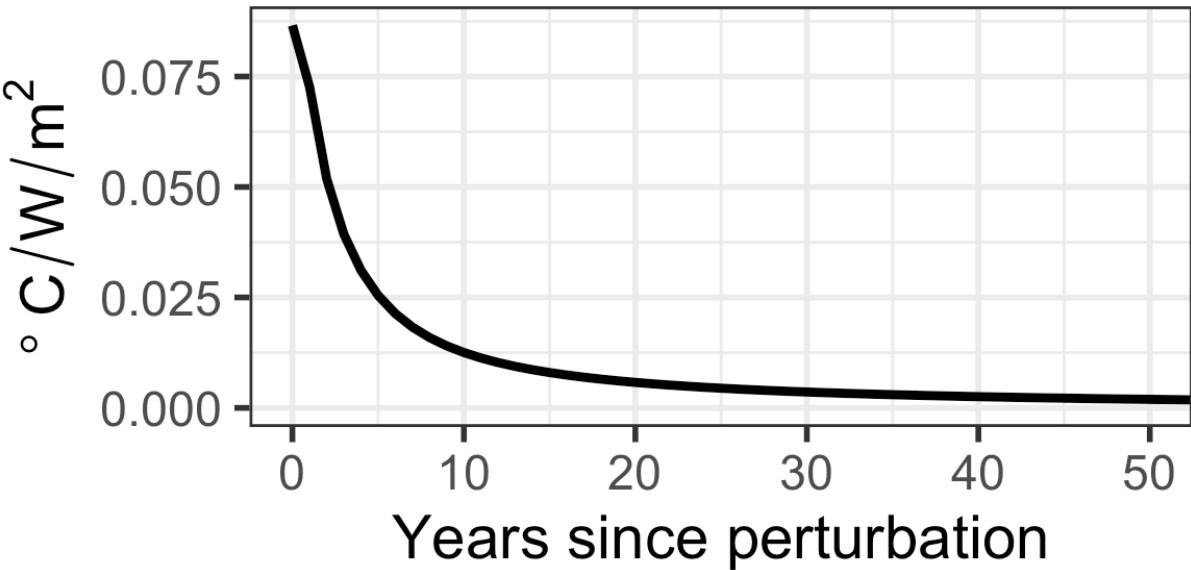

**Figure 1: The first 50 years of the global temperature response to a radiative forcing perturbation for Hector v2.0; the remaining 2,500 years of the impulse response are almost constant and slowly approach zero. Here the black carbon emissions were doubled in 2010 relative to the Representative Concentration Pathway 4.5 value.**

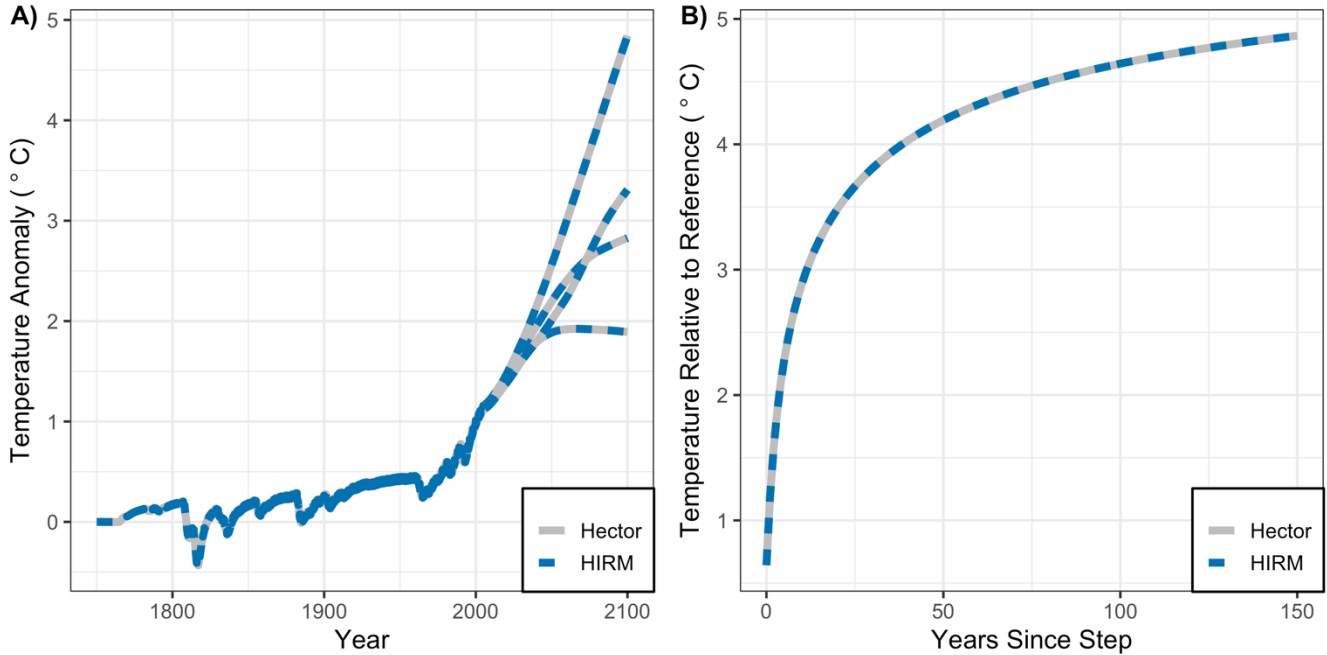


**Figure 2: Comparison of Hector (grey dashed) and HIRM (blue dashed) global mean temperature anomaly from the two validation experiments. In panel (A) HIRM was used to the recreate Hector temperature for the four RCPs. The four lines in panel (A) from lowest to highest 2100 temperature represent results for RCP 2.6, RCP 4.5, RCP 6.0 and RCP 8.5. Panel (B) compares the temperature response of HIRM and Hector from the abrupt four times $CO_2$ concentration step validation test.**


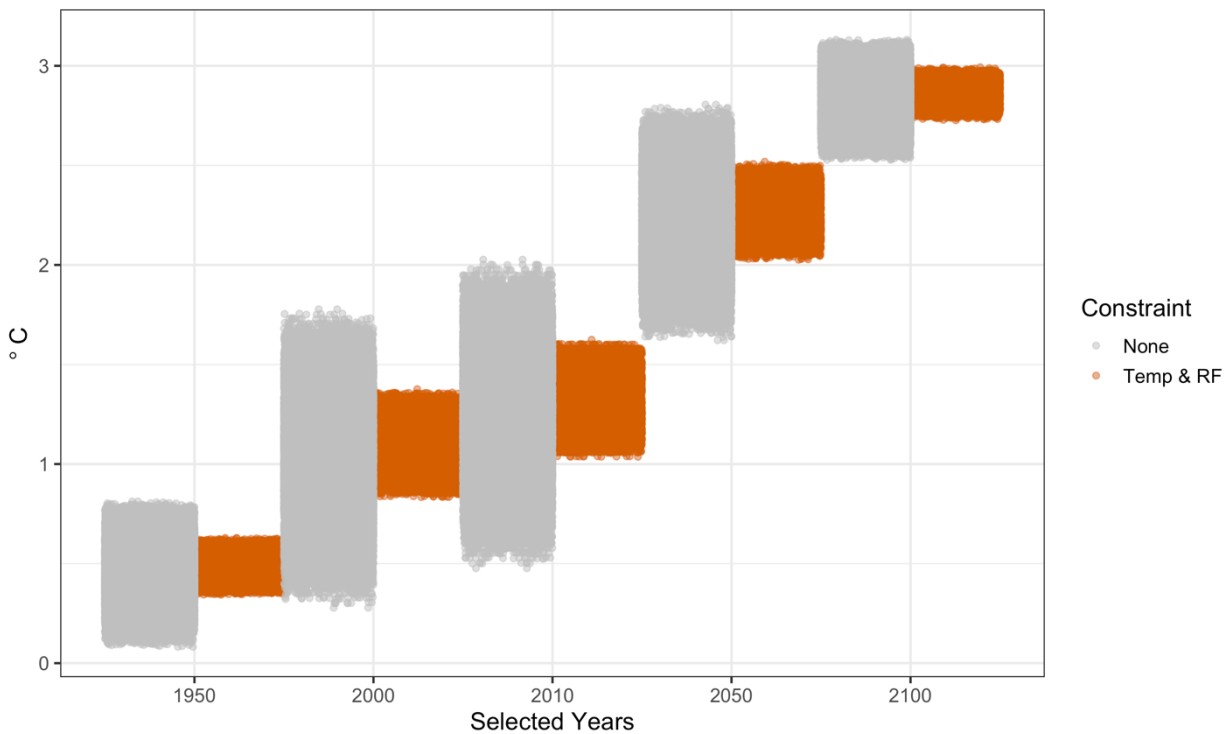

**Figure 3: The temperature (°C)_spread from the aerosol uncertainty runs in selected years. The grey regions show all of the possible runs before the historical constraints were put into the place; orange regions are the runs that passed through both historical temperature and radiative forcing constraints. The uncertainty in temperature due to uncertainty in aerosol forcing decreases by 2100 because emissions of aerosols and precursor compounds decrease over time so their influence on temperature decays over time as well. We note that uncertainty in other climate system parameters, such as climate sensitivity and ocean heat diffusivity, were not samples in this application. Including these uncertainties would alter these results. Note that temperature change in 2020 is larger than the applied historical constraint ( [0.65 to 1.1] °C over 1880–2012) because temperatures in this figure are relative to 1750.**


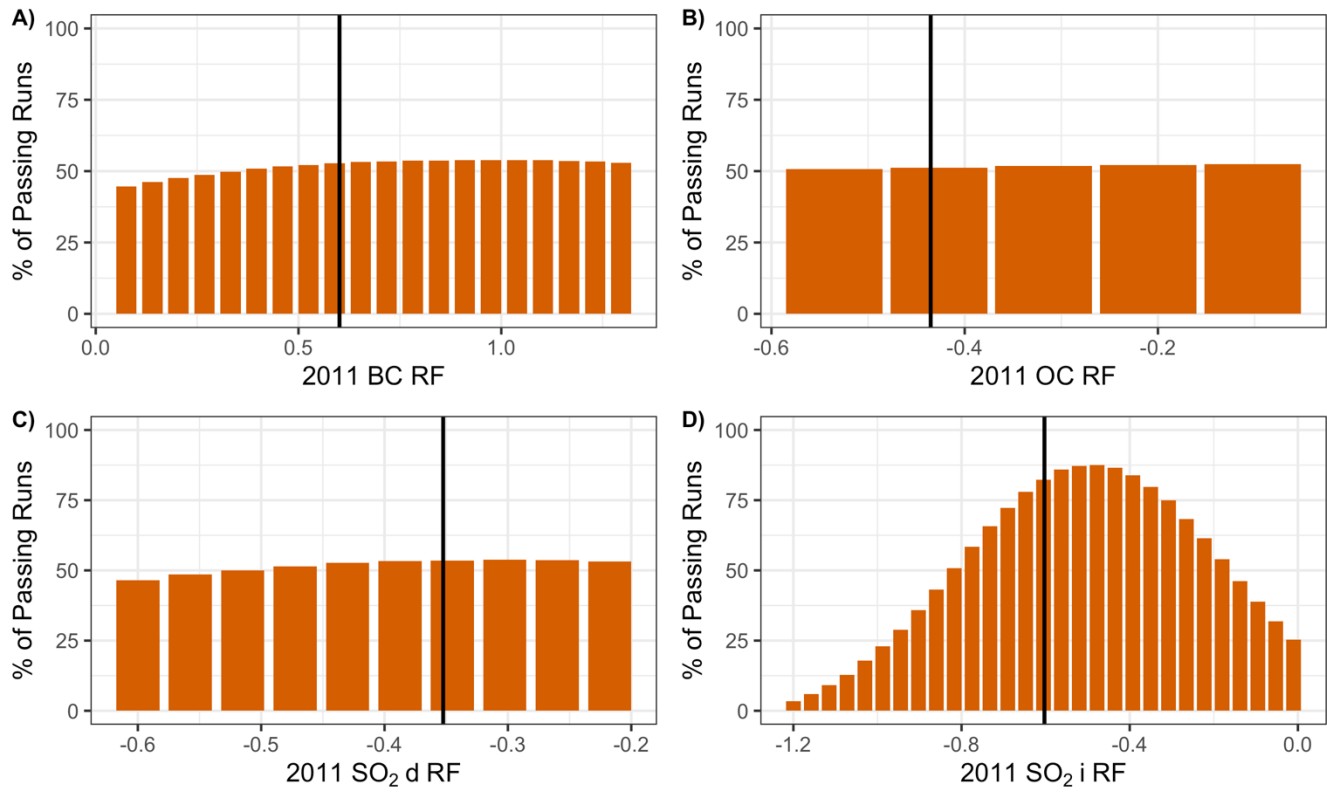


**Figure 4: Uncertainty scalers used to vary (A) black carbon, (B) organic carbon, (C) direct SO₂ effects, and (D) indirect SO₂ effects aerosol RF time series in the uncertainty analysis. HIRM was run a total of 29000 times with every combination of uncertainty scaler represented on the x-axes of panels A-D, creating an ensemble of uncertainty runs with scalars varying for all radiative forcing agents. Each panel of this figure plots a projection of the percent of runs passing through the historical constraints as the 2011 radiative forcing agent of an agent is varied. The black vertical line marks default 2011 RF.**


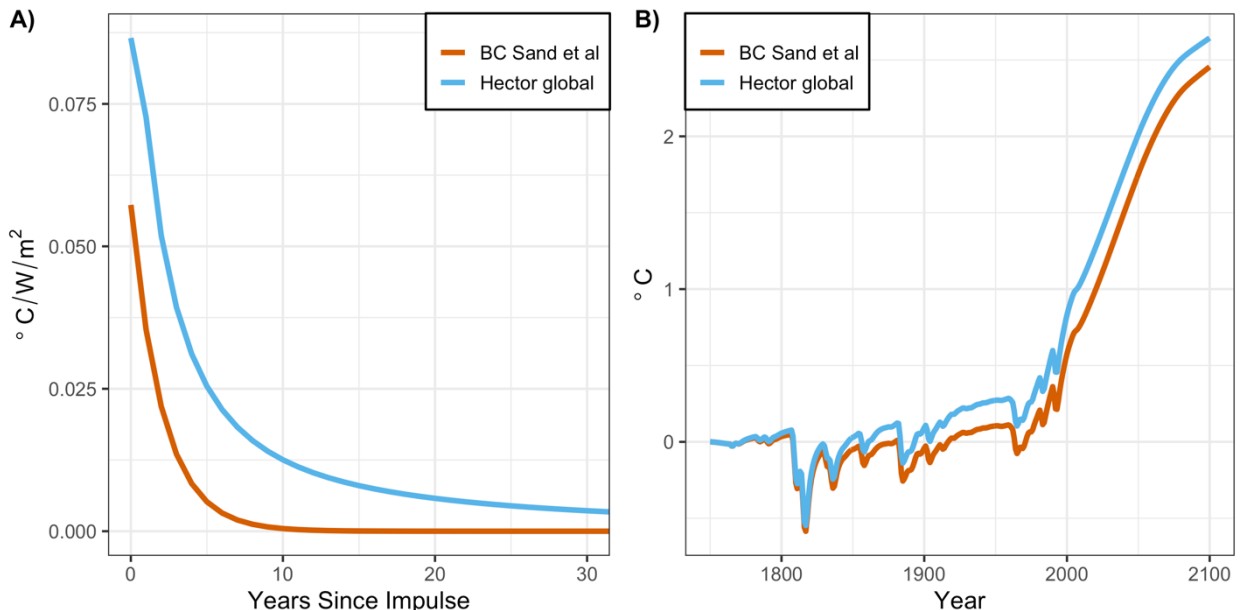

**Figure 5: (A) Hector's IRF (blue) compared with the BC Sand et al. 2015 IRF (red). (B) HIRM total temperature for the Representative Concentration Pathway 4.5 for two HIRM cases, one that only uses Hector's IRF (blue) and the other pairing the BC RF time series with the BC Sand et al. 2015 IRF (red).**
