# Peer review of "HIRM v1.0: A hybrid impulse response model for climate modeling and uncertainty analyses"

_Geoscientific Model Development, 2020_

## Referee Comment (RC1) · Ben Kravitz (Referee) · 1 May 2020

The authors present a really interesting approach to simple climate modeling that I hadn't considered before. I think this is a smart idea. The authors have done a thorough job with their analysis. My only comments are related to phrasing and context of the results. I am recommending minor revisions.

Lines 12-14: This sentence caught my attention as needing more caveating. In linear time-invariant systems the impulse response fully characterizes system dynamics. In nonlinear systems it doesn't. So as written, this sentence is coming across as though your fundamental methodology is flawed (which I'm sure it's not – based on my reading of lines 51-52, you do understand the distinction). I do agree that your approach captures _some_ of the dynamics, perhaps even the most important parts of it, depending on what dynamics your model is designed to represent. Some rephrasing is needed.

Lines 32-34: That's one way to do it. There are others, for example: https://agupubs.onlinelibrary.wiley.com/doi/abs/10.1029/2011GL048623

Line 39: I'd be careful with the word "nonlinearities". You can have linear feedbacks that still result in interesting dynamical behavior. I agree that chemistry can result in substantial nonlinearities (I think Kate Marvel had a paper on this looking at single forcings), and some GHGs are known to have a nonlinear relationship between concentration and forcing, but talking about the entire suite of carbon cycle feedbacks as nonlinear is perhaps too much.

Lines 44-45: Again, they _can_ be. They don't have to be.

Lines 68-72: Clever. And an excellent description of where the potential problems lie. And I appreciate the validation of your assumptions later.

Lines 89-90: This assumption is known to be incorrect (i.e., efficacy; Hansen et al., 2005). That's not a problem for your analysis, but you'll need a caveat on your interpretation of your results.

Lines 96-97: This strikes me as appropriate. It's always tricky to delineate temperature from response because they coevolve, but your choice here makes sense.

Line 110: How do you impose the RF pulse in the model? (Or if this is described later, say so.)

Section 2.3 is a bit odd. I expected to actually see the results of the validation here. Perhaps move these short descriptions to more relevant points of the manuscript?

Lines 132-133: I'm having trouble understanding this sentence.

Section 4.1: I'm having a bit of trouble understanding exactly what you did. If I understand it correctly, you (1) come up with ranges of uncertainties for each of those aerosol

forcing terms, (2) sample those spaces to come up with 29,000 sets of parameters that you call uncertainty scalers, and (3) simulate those combinations in HIRM, throwing out results that don't match historical radiative forcing and temperature?

Line 214: Typo

Line 297: Typo in your acronym

Figure 2: Can you choose different colors? Orange and gray are difficult on the eyes.

---

## Referee Comment (RC2) · Kuno Strassmann (Referee) · 29 May 2020

General comments

1. The manuscript describes the emulation of the temperature response to radiative forcing of a simple climate model with an impulse-response function. This is not a novel concept, and corresponds to what is typically one out of a set of many equations in existing simple climate models. The wording used by the authors suggests a more sophisticated, original approach, and is in my opinion a bit misleading. My main objection, however, is that the limitations of the model, which are quite strong, are not sufficiently addressed. The main limitation is that the (climate) model is not really coupled to a carbon cycle component and cannot therefore represent the carbon-cycle-climate-

feedback. Another important but never mentioned limitation is that the model presented has a fixed climate sensitivity. The approach could be extended by extracting IRFs from models (or model setups) with different sensitivities, but this is not straightforward.

2. Furthermore, I find the validation exercises presented (section 2.3) not very informative, and I feel that the case studies (section 4) are only useful to demonstrate model application, and do not yield significant scientific insight. Several of the conclusions offered based on the latter appear unfounded.

3. Finally, the model description is incomplete, as no details on implementation are given, short of the model code itself. This said, the model as such seems to be correct, and my comments only concern its presentation, applications, and interpretation. I leave it to the editor to decide whether the limited material presented here warrants a publication in gmd.

Specific comments

1. p1/25 Earth system models of intermediate complexity should be mentioned here, especially as they are referred to later in the paper.

2. p2/31 "SCMs can be characterized as either process-based or idealized climate models." All models, especially SCMs are idealized - a better word would be "abstract" as opposed to "process-based". However, it would be better to speak of IRF-based models, as the authors use the term "idealized SCM" synonymously, although other types of non-process-based models may exist.

3. p2/40 Another key nonlinearity that should be mentioned here is the chemistry of $CO_2$ uptake at the ocean surface.

4. p2/44 Are all idealized SCMs based on IRFs? It would be better (and avoid repetition) to say "IRFs used in SCMs are defined as. . .".

5. p2/48 AR5 mentions several types of IRF-model, but the main model used to calculate GWP represents the relationship between emissions and $CO_2$, not temperature.

6. p2/53 "Idealized SCMs may exhibit biased results, however, due to their lack of nonlinear dynamics." It is not in general true that idealized SCMs, meaning SCMs that are not (fully) process-based and use IRF functions, lack nonlinear dynamics. There are SCMs that apply IRFs only to the quasi-linear parts of the system, linking these with equations that capture essential nonlinearities (e.g. Joos and Bruno, 1996; Strassmann and Joos, 2018).

7. p2/56 Using the RF simulated by a model with nonlinearities as input hardly counts as "incorporating nonlinear dynamics".

8. p3/65 (whole paragraph) I find it rather misleading to call the use of an existing model to provide input a "framework", given that there does not seem to be any real coupling, i.e. exchange of information at intermediate timesteps. If this is the characteristic that distinguishes this "hybrid approach" from other IRF-based models, it does so in a negative way. There are SCMs that represent the climate response with IRFs and allow for coupling with a carbon cycle component at each timestep, for example, the BernSCM model (Strassmann and Joos, 2018). BernSCM combines IRF-based components describing linear systems with nonlinear parametrisations to capture the essential nonlinearities of the carbon cycle-climate system, and expresses the IRF-components as a system of ordinary differential equations to allow for efficient integration in coupled mode.

9. p3/69 "incorporate the nonlinear dynamics... if the majority of the nonlinear dynamics of SCMs occur between the emissions to radiative forcing calculation" -it would be more adequate here to say that the IRF-model, which really constitutes the contribution of the authors, does NOT incorporate any nonlinearities, because there aren't any.

10. p4/96 (whole paragraph) It is true that the carbon-cycle-climate feedback could be included in the IRF. However, the resulting model would still have strong limitations. It is likely that such a model, being liearized, would give accurate results only for a limited range of forcing scenarios or time scales.

11. p4/114 "The end of the IRF was extrapolated with an exponential decay function" Please mention the decay timescale.

12. p4/120 "underlying assumptions about where the majority of the nonlinearities occur are true" - This simply means the climate component of Hector is linear, which is to be expected of an SCM and could be inferred by looking at the design of that model.

13. p4/122 I don't see what chemical processes could affect the relationship between RF and temperature, at least in an SCM.

14. p5/126 "For each RCP..." this sentence is not very informative and could be dropped.

15. p5/149 As mentioned above, this finding is not surprising; it merely characterizes the Hector SCM and holds no scientific information on the physical climate as such.

16. p5/155 "difference of 0.0%" There are no significant digits in this number.

17. p6/156 As for the additivity of temperature changes, the lack of nonlinearities in the Hector climate component is not a scientifically relevant finding.

18. p6/164 "In this analysis, however, the black carbon (BC), organic carbon (OC), indirect SO2 effects (SO2i), and direct SO2 effects (SO2d) RF input time series were varied." It is not correct to only vary these components. The uncertainty of other forcings should also be considered. The uncertainty of CO2 RF, for example, though small in relative terms, is important due to the dominant contribution of that component. Leaving out these uncertainties will result in an overconstrained temperature range.

19. p6/169 "sampled at intervals of 0.04 W/m2" It should perhaps be mentioned that this sampling does not produce a plausible probabilistic distribution of the results, since the RF uncertainties cannot be assumed to be uniformly distributed. Since the authors do not make a probabilistic interpretation, this is not a major issue, however.

20. p7/189 "This shows..." due to the overconstraining mentioned, this result is not

valid in my opinion. Consequently the excercise described is only useful as an illustration of using the model framework, as stated in the following sentence.

21. p7/193 (section 4.1) It is possible to use an IRF for a specific component from another model, as the authors do, but I am not sure how meaningful this is, since this mixes the climate responses of two different models. To get a consistent model emulation the IRFs for the other RF components should, in principle, be taken from the same model (i.e., NorESM-1).

22. p8/239 "it demonstrates nonlinear dynamics" I find this claim unfounded since the nonlinearities in question concern the dynamics of a previously existing model, while the model component contributed by the authors cannot represent the relevant nonlinearity, i.e., that of the climate-carbon cycle feedback.

23. p9/253 (whole paragraph) "most of the linearities" - there is no finding about any specific nonlinearities, and the fact that the Hector SCM has a linear RF-temperature response is no basis for a recommendation for further model development. The near-linear relationship between RF and temperature is well known and has been demonstrated and exploited in SCMs for a long time (e.g., Joos and Bruno, 1996).

24. p9/271 "we demonstrate that the use of a forcing-based impulse response function overcomes most of these limitations." I don't see that any limitations are overcome by this approach.

25. p9/273 "These findings imply…" Again, there is no basis for such a recommendation.

Technical corrections

- p5/132 "In this experiment HIRM was configured" - The word "was" seems to be superfluous here. - In Table 1, the unit should be given.

References

- Joos and Bruno, 1996. Pulse response functions are cost-efficient tools to model the link between carbon emissions, atmospheric CO2 and global warming, Phys. Chem. Earth, 21, 471–476.

- Strassmann, K. M. and Joos, F.: The Bern Simple Climate Model (BernSCM) v1.0: an extensible and fully documented open-source re-implementation of the Bern reduced-form model for global carbon cycle–climate simulations, Geosci. Model Dev., 11, 1887–1908, https://doi.org/10.5194/gmd-11-1887-2018, 2018.

---

## Referee Comment (RC3) · Anonymous Referee #3 · 1 Jul 2020

The authors present a model that couples radiative forcing (potentially from any source, but here solely from the Hector simple climate model) to an impulse response function to calculate global mean surface temperature anomaly, with the possibility to choose different impulse response functions for different forcings (e.g. black carbon).

It is difficult to see what kind of advance this model is. HIRM relies very heavily on Hector, the details of which are documented extensively elsewhere. Using different IRFs for different forcings is not a new concept either (e.g. Richardson et al 2019 for CO2, CH4, BC, SO2 and solar forcing, Larson and Portmann 2016 for volcanic as a special case).

This could be a nice module to include in Hector as an alternative way to calculate temperature in that model. The comparison to the default Hector temperature response

function is shown in Figure 2 and seen to be almost identical, so this simplification (is it a simplification?) may be worthwhile. But, due to its nearly total dependence on Hector and the fact that species-dependent efficiacy response functions have been done previously, it doesn't qualify as a brand new model for me. Rather it is a submodule of Hector. It is possible that there is more to this paper than meets the eye, but if there is it should stand out more, and if the authors believe this does warrant a standalone model, expend some effort in decoupling it from Hector and explain what is improved or new over e.g. Richardson et al. 2019.

Specific comments:

Line 18: Examining the effect of aerosol forcing on global temperature: a worthwhile cause. There is not actually done in this paper however. To me this would involve varying the present-day forcing of aerosols, climate sensitivity, and carbon cycle feedbacks, and investigating how this would cause temperature to evolve in a probabilistic fashion. The projections shown in figure 3 are far too constrained as explained in a later comment.

Around line 25, there is a missing link between ESMs and SCMs: Earth System Models of Intermediate Complexity (EMICs). In fact, you could say that in decreasing order of complexity we have ESMs > GCMs > EMICs > SCMs.

In the paragraph starting on line 32, the authors discuss the differences between process-based and idealized simple climate models, presumably as a preface for introducing their own model that couples the two components. It is not clear to me that these two concepts are necessarily separate, and if they are, this model may not be as novel as the authors claim. The later versions of FAIR (Smith et al., 2018, GRL) include an impulse response function for CO2 emissions to concentrations and for converting forcing to temperature, and "processed-based" representations of concentrations of greenhouse gases, radiative forcing of GHGs and short-lived climate forcers, and feedbacks from temperature on the carbon cycle and radiative forcing. Leach et al. (2020)

in the Generalised Impulse Response model extends the impulse-response framework of the carbon cycle to other greenhouse gases and short-lived climate forcers. In both of these models, the radiative forcing is internally calculated (process-based, in the language of the authors?) and not supplied externally/provided by a different model (as discussed in lines 82-83 for HIRM). For my benefit if not others, could you cite maybe one example of a "process-based" SCM on line 32 if these concepts are separate? MAGICC perhaps?

line 37: "top of troposphere" - this is an old and incomplete definition of radiative forcing, and effective radiative forcing is now preferred - the Smith et al (2018, JGR) reference which is in the bibliography (but not in the text, oddly) goes into some detail on this. I should say this discussion is of limited importance for SCMs.

line 54: "in addition, the physical interpretation of their behaviour is not always straightforward". Please explain why not.

Line 90: This is a confusing paragraph. On first reading it seems like HIRM doesn't allow for species-dependent efficacy. Then I later read the discussion on BC, and see that the different IRF for BC can be included, which *is* a different efficacy (and time-dependent too). Then on second reading I see that the authors are talking about Hector not having species-dependent efficacy which is more evidence that this model is a component of Hector and not standalone. In general, in section 2.2 it is difficult to follow what the authors did. A flow diagram could help.

Lines 97-102: It is not correct to exclude carbon cycle feedbacks. It is no good if HIRM can emulate Hector with feedbacks switched off if the latter is not representative of the real world. If the forcing comes from Hector in the first place, feedbacks need not be excluded in the Hector configuration, because HIRM doesn't calculate forcing. The analogy here would be concentration-driven GCM runs, where the concentrations are calculated by MAGICC (which includes carbon cycle feedbacks) but the GCMs themselves do not, going from concentrations > forcing > temperature.

[Figure]

line 120-121: people have underlying assumptions, but software doesn't

line 124: Hector's IRF - this is the function in figure 1 isn't it, because Hector is not impulse-response based. Could just refer to confirm.

lines 131-132: it goes without saying that 4xCO2 tests the climate model's response to CO2. The importance of the 4xCO2 experiment is that it can be used to (imperfectly) estimate climate sensitivity, climate feedback and CO2 radiative forcing in GCMs (Gregory et al 2004) by using a forcing with a high enough signal-to-noise ratio to get a clear signal but still small enough to avoid substantial non-linearities and tipping points. Hence it can be used as a line of evidence for climate sensitivity, which itself is an input parameter to many simple climate models. Also, putting the Schwarber reference in line 132 reads like they invented this experiment.

line 145: The difference ... and the following sentence, can be dropped. It's apparent from the naked eye that the differences are imperceptible, I don't think this needs to be rigorous. Similar sentence in following paragraph.

line 157: Which SCM? Hector?

lines 161-162: needs a reference

line 174: 29000 is a bit of a random number, is there a motivation for this?

lines 176-177: wrong values (-1.9 to -0.1 is AR5 "very likely" i.e. 5-95% estimate), and also wrong citation (Myhre et al., 2013).

Figure 3: why does uncertainty reduce over time? 2100 temperature is very tightly constrained, but this is the timeframe over which uncertainties in radiative forcing, climate sensitivity and carbon cycle feedbacks multiply. I know these are not sampled, but this should very clearly be stated and the fact that this is not a true future uncertainty quantification of warming. I'd also check the constraints - is 1.6C of warming in 2010, which passes the constraint, realistic?

lines 189-190: Important point, long known. Compare/cite some relevant studies e.g. Forest (2018). Figure 4 would be more naturally expressed in terms of a W/m2 aerosol forcing posterior for e.g year 2010, perhaps add a subpanel e. This would back up the claim that strong aerosol forcing values do not pass the constraints.

Line 214: Mention the perturbation size from line 220 here. Richardson et al. (2019) included a multi-model response for BC and would be more appropriate to use than the single-model study here.

line 230: maybe a better wording would be "... the global temperature was 0.2C lower using the specific BC IRF from Sand et al. (2015)." Avoid using "decreased" in this sentence.

minor:

line 9: Earth (rather than earth)

line 29-30: would probably get picked up in proofing but check citation spellings (Meehl, Meinshausen)

line 40: Myhre et al.

line 91: forging

line 200: also Richardson et al. 2019

line 226: units, should be C/(W/m2)? i.e. the m2 is in the numerator

line 253: HRIM (and in several other places)

line 244: "a" not required

line 245: "dynamics" - not really dynamics is it - just a different IRF.

line 438: 29000 times

References:

Richardson et al. 2019: https://agupubs.onlinelibrary.wiley.com/doi/full/10.1029/2019JD030581

Larson and Portmann 2016: https://journals.ametsoc.org/jcli/article/29/4/1497/35504/A-Temporal-Kernel-Method-to-Compute-Effective

Smith et al. 2018, GRL: https://gmd.copernicus.org/articles/11/2273/2018/

Leach et al. 2020: https://gmd.copernicus.org/preprints/gmd-2019-379/

Forster et al. 2016: https://agupubs.onlinelibrary.wiley.com/doi/full/10.1002/2016JD025320

Smith et al. 2018, JGR: https://agupubs.onlinelibrary.wiley.com/doi/full/10.1029/2018GL079826

Gregory et al. 2004: https://agupubs.onlinelibrary.wiley.com/doi/10.1029/2003GL018747

Forest 2018: https://link.springer.com/article/10.1007/s40641-018-0085-2

---

## Author Comment (AC1) · 2 Sep 2020

Response to Referees
gmd-2020-33
Reviewer comments in normal type face. **Response in bold.**

Ben Kravitz (Referee #1)

The authors present a really interesting approach to simple climate modeling that I hadn't considered before. I think this is a smart idea. The authors have done a thorough job with their analysis. My only comments are related to phrasing and context of the results. I am recommending minor revisions.

**Thank you for taking the time to review this manuscript. We believe that our changes address the issues raised in your review.**

Lines 12-14: This sentence caught my attention as needing more caveating. In linear time-invariant systems the impulse response fully characterizes system dynamics. In nonlinear systems it doesn't. So as written, this sentence is coming across as though your fundamental methodology is flawed (which I'm sure it's not – based on my reading of lines 51-52, you do understand the distinction). I do agree that your approach captures _some_ of the dynamics, perhaps even the most important parts of it, depending on what dynamics your model is designed to represent. Some rephrasing is needed.

**We have revised this sentence to read:**
**This structure allows it to capture the most crucial nonlinear dynamics encountered in going from greenhouse gas emissions to atmospheric concentration to radiative forcing.**

Lines 32-34: That's one way to do it. There are others, for example:
https://agupubs.onlinelibrary.wiley.com/doi/abs/10.1029/2011GL048623

**Thank you for bringing this publication to our attention. This is an interesting analysis in frequency space, but we're not sure how this would be translated into the applications to which SCMs are commonly applied, such as scenario analysis. We changed the text to clarify there are multiple ways to approach idealized simple climate modeling, see paragraph starting at line 33.**

Line 39: I'd be careful with the word "nonlinearities". You can have linear feedbacks that still result in interesting dynamical behavior. I agree that chemistry can result in substantial nonlinearities (I think Kate Marvel had a paper on this looking at single forcings), and some GHGs are known to have a nonlinear relationship between concentration and forcing, but talking about the entire suite of carbon cycle feedbacks as nonlinear is perhaps too much.

**Agreed, we adjusted the text in lines 47-51 to improve our discussion about how process based models incorporate complex climate interactions and nonlinear climate dynamics.**

Lines 44-45: Again, they _can_ be. They don't have to be.

**Good point. We borrow this terminology from Millar et al. 2017 and have adjusted the text starting in line 33 to better reflect this and caveat that there are multiple ways to use IRFs in simple climate modeling.**

Lines 68-72: Clever. And an excellent description of where the potential problems lie. And I appreciate the validation of your assumptions later.

**Thank you!**

Lines 89-90: This assumption is known to be incorrect (i.e., efficacy; Hansen et al., 2005). That's not a problem for your analysis, but you'll need a caveat on your interpretation of your results.

**Good point, this was not properly conveyed in the previous draft of the manuscript. We've added text in lines 86-88, and 106-109 to more clearly communicate this point.**

Lines 96-97: This strikes me as appropriate. It's always tricky to delineate temperature from response because they coevolve, but your choice here makes sense.
**Thank you.**

Line 110: How do you impose the RF pulse in the model? (Or if this is described later, say so.)

**The RF pulse is equal to the difference between the radiative forcing between the reference and the BC emission perturbation run. Text in lines 132 and 139 was modified to clarify this point.**

Section 2.3 is a bit odd. I expected to actually see the results of the validation here. Perhaps move these short descriptions to more relevant points of the manuscript?

**Section 2 and 3 were combined/reorganized so that the results of the validation tests were paired with the description of the test setup.**

Lines 132-133: I'm having trouble understanding this sentence.

**The sentence now reads "In this experiment HIRM was set up with the Hector derived IRF and a RF input from an abrupt four times CO2 concentration step".**

Section 4.1: I'm having a bit of trouble understanding exactly what you did. If I understand it correctly, you (1) come up with ranges of uncertainties for each of those aerosol forcing terms, (2) sample those spaces to come up with 29,000 sets of parameters that you call uncertainty scalers, and (3) simulate those combinations in HIRM, throwing out results that don't match historical radiative forcing and temperature?

**Yes, that is what we did. The wording in this section was changed to more clearly explain this case study, please see lines 192-227.**

Line 214: Typo
**Corrected.**

Line 297: Typo in your acronym
**Corrected.**

Figure 2: Can you choose different colors? Orange and gray are difficult on the eyes.
**We've selected another grey-color combination from a color-blind friendly color palette.**

---

## Author Comment (AC2) · 2 Sep 2020

Response to Referees
gmd-2020-33
Reviewer comments in normal type face. **Response in bold.**
* * *
**Thank you for taking the time to review this manuscript. We value your feedback and believe that it has led to better, clearer version of the manuscript.**

General comments

1. The manuscript describes the emulation of the temperature response to radiative forcing of a simple climate model with an impulse-response function. This is not a novel concept, and corresponds to what is typically one out of a set of many equations in existing simple climate models. The wording used by the authors suggests a more sophisticated, original approach, and is in my opinion a bit misleading. My main objection, however, is that the limitations of the model, which are quite strong, are not sufficiently addressed. The main limitation is that the (climate) model is not really coupled to a carbon cycle component and cannot therefore represent the carbon-cycle-climate feedback. Another important but never mentioned limitation is that the model presented has a fixed climate sensitivity. The approach could be extended by extracting IRFs from models (or model setups) with different sensitivities, but this is not straightforward.

**We do believe that this is a new method to incorporate these dynamics into IRF based SCMs (we do not know of other modeling groups that do it this way) but by no means did we intend to suggest that this is the only way it has been done. We've expanded text in the introduction to make it clear to readers that there are several ways to approach this.**

**We've also added a new section 2.1 Parent Model Description (starting on line 66) to provide a clearer description of the connection between HIRM and Hector; we believe that this addresses your concerns related to the carbon cycle and limitations of the model. Model limitations are also discussed in lines (307-313).**

**Thank you for pointing out that HIRM could be configured with different climate sensitivities, that is a great idea and we have incorporated it as a future area of research in the discussion section (lines 344 - 346).**

2. Furthermore, I find the validation exercises presented (section 2.3) not very informative, and I feel that the case studies (section 4) are only useful to demonstrate model application, and do not yield significant scientific insight. Several of the conclusions offered based on the latter appear unfounded.

**We appreciate the Reviewer's point of view, but the case studies were chosen precisely to "demonstrate model application"--this is Geoscientific Model Development after all, a journal founded in recognition that model description papers are significant and necessary science. That said, we have made a number of modifications to section 4 that we feel have put into better context both of the validation and case studies.**

3. Finally, the model description is incomplete, as no details on implementation are given, short of the model code itself. This said, the model as such seems to be correct, and my comments only concern its presentation, applications, and interpretation. I leave it to the editor to decide whether the limited material presented here warrants a publication in gmd.

**Because Hector, the SCM model used here, is described in a number of previous publications, we have expanded short summary of its structure and capabilities and provide citations for readers interested in more detail (see section 2.1). This allows the current manuscript to focus on HIRM and its applications. With respect to HIRM itself, a new paragraph (lines 110 - 114) describes its structure and implementation details.**

Specific comments

1. p1/25 Earth system models of intermediate complexity should be mentioned here, especially as they are referred to later in the paper.

**EMICs have been incorporated into the manuscript starting in line 26.**

2. p2/31 "SCMs can be characterized as either process-based or idealized climate models." All models, especially SCMs are idealized - a better word would be "abstract" as opposed to "process-based". However, it would be better to speak of IRF-based models, as the authors use the term "idealized SCM" synonymously, although other types of non-process-based models may exist.

**This is an interesting point. We adopted this terminology from Millar et al. 2017 to be consistent with the language used by other simple climate modeling groups. This being said we hope the changes made to the introduction and in lines 35-39 address these concerns.**

3. p2/40 Another key nonlinearity that should be mentioned here is the chemistry of CO2 uptake at the ocean surface.

**Thanks for this suggestion, as it is an important nonlinearity. We have added it to line 50.**

4. p2/44 Are all idealized SCMs based on IRFs? It would be better (and avoid repetition) to say "IRFs used in SCMs are defined as. . .".

**This sentence has been removed.**

5. p2/48 AR5 mentions several types of IRF-model, but the main model used to calculate GWP represents the relationship between emissions and CO2, not temperature.

**Thank you for pointing out the inaccuracy here. Due to changes in this section, this sentence has been removed.**

6. p2/53 "ldealized SCMs may exhibit biased results, however, due to their lack of nonlinear dynamics." It is not in general true that idealized SCMs, meaning SCMs that are not (fully) process-based and use IRF functions, lack nonlinear dynamics. There are SCMs that apply IRFs only to the quasi-linear parts of the system, linking these with equations that capture essential nonlinearities (e.g. Joos and Bruno, 1996; Strassmann and Joos, 2018).

**This was a miscommunication on our part; please see (lines 37-40) for the modified text which better situates HIRM among other nonlinear IRF based climate models. Thank you for sharing these references–they have been incorporated into the manuscript.**

7. p2/56 Using the RF simulated by a model with nonlinearities as input hardly counts as "incorporating nonlinear dynamics".

**We've tweaked the wording of the sentence so that now is reads: "In this manuscript we document and demonstrate a new highly idealized IRF-based framework".**

8. p3/65 (whole paragraph) I find it rather misleading to call the use of an existing model to provide input a "framework", given that there does not seem to be any real coupling, i.e. exchange of information at intermediate timesteps. If this is the characteristic that distinguishes this "hybrid approach" from other IRF-based models, it does so in a negative way. There are SCMs that represent the climate response with IRFs and allow for coupling with a carbon cycle component at each timestep, for example, the BernSCM model (Strassmann and Joos, 2018). BernSCM combines IRF-based components describing linear systems with nonlinear parametrisations to capture the essential nonlinearities of the carbon cycle-climate system, and expresses the IRF-components as a system of ordinary differential equations to allow for efficient integration in coupled mode.

**This paragraph has been edited to clarify how HIRM can be used with more sophisticated process-based models. However, we do feel that the term framework is sufficiently general to cover the application we have described here.**

9. p3/69 "incorporate the nonlinear dynamics. . . if the majority of the nonlinear dynamics of SCMs occur between the emissions to radiative forcing calculation" -it would be more adequate here to say that the IRF-model, which really constitutes the contribution of the authors, does NOT incorporate any nonlinearities, because there aren't any

**Noted and this sentence has been removed as part of edits to clarify the presentation in response to referee comments. We aimed for the revised text to communicate that the non-linear dynamics are, indeed, incorporated only through the use of forcing time series from the parent SCM.**

10. p4/96 (whole paragraph) It is true that the carbon-cycle-climate feedback could be included in the IRF. However, the resulting model would still have strong limitations. It is likely that such a model, being liearized, would give accurate results only for a limited range of forcing scenarios or time scales.

**We agree, and mention this in the paper (see lines 123-129).**

11. p4/114 "The end of the IRF was extrapolated with an exponential decay function" Please mention the decay timescale.

**More details about the exponential decay have been added: line (143-144) now reads "This IRF had a length of 300 years, in order to ensure the IRF was long enough to be convolved with the RF inputs; the end of the IRF was extrapolated with an exponential decay function to a length of 3000 with a decay constant of 0.20".**

12. p4/120 "underlying assumptions about where the majority of the nonlinearities occur are true" - This simply means the climate component of Hector is linear, which is to be expected of an SCM and could be inferred by looking at the design of that model.

**Hector incorporates a diffusive model for ocean heat uptake which is, by definition, non-linear. However, as we have shown here, this nonlinearity is relatively weak compared to the non-linearities in emissions -> forcing calculations.**

13. p4/122 I don't see what chemical processes could affect the relationship between RF and temperature, at least in an SCM.

**We have revised the text in lines (154-156) to clarify. (Chemical processes can impact relationships between emissions and forcing.)**

14. p5/126 "For each RCP. . ." this sentence is not very informative and could be dropped.

**We believe that this text does provide important information about the validation test set up, however we have modified the text so that information is more clearly portrayed.**

15. p5/149 As mentioned above, this finding is not surprising; it merely characterizes the Hector SCM and holds no scientific information on the physical climate as such.

**This sentence is not intended to convey any information on the physical climate system but is intended to show the extent to which the HIRM emulation matches the original hector result. As noted above, such a good match is not a foregone conclusion.**

16. p5/155 "difference of 0.0%" There are no significant digits in this number

**Noted, please see line 172 where we have updated this to 0%,**

17. p6/156 As for the additivity of temperature changes, the lack of nonlinearities in the Hector climate component is not a scientifically relevant finding.

**As noted above, this is not a foregone conclusion.**

18. p6/164 "In this analysis, however, the black carbon (BC), organic carbon (OC), indirect SO2 effects (SO2i), and direct SO2 effects (SO2d) RF input time series were varied." It is not correct to only vary these components. The uncertainty of other forcings should also be considered. The uncertainty of CO2 RF, for example, though small in relative terms, is important due to the dominant contribution of that component. Leaving out these uncertainties will result in an overconstrained temperature range.

**We certainly agree with the comments of the reviewer in terms of scientific principles. We note, however, that the purpose of this section is to demonstrate how the tool could be used (as part of the GMD model documentation paper), not to conduct a rigorous uncertainty analysis.**

**This has been noted in the modified text.**

19. p6/169 "sampled at intervals of 0.04 W/m2" It should perhaps be mentioned that this sampling does not produce a plausible probabilistic distribution of the results, since the RF uncertainties cannot be assumed to be uniformly distributed. Since the authors do not make a probabilistic interpretation, this is not a major issue, however

**Correct we do not produce probabilistic results and have added text in 1997 to explicitly state that this case study does not produce probabilistic results.**

20. p7/189 "This shows. . ." due to the overconstraining mentioned, this result is not valid in my opinion. Consequently the exercise described is only useful as an illustration of using the model framework, as stated in the following sentence.

**This is, indeed, an application of the model as noted. However, we disagree that this conclusion is not valid. We note that most historical aerosol forcing estimation experiments don't separately examine the effects of different aerosol components (e.g., Forest 2018) - although Tomassini et al 2007 and Meinshausen et al. 2009 are exceptions and these citations have been added (although numerical values are not available making it difficult for us to compare directly). Most of these experiments generally scale total aerosol forcing up or down. In this exercise we have full forcing time series with different time paths for each forcing component (see Smith and Bond 2014 for examples). So we believe that our statement of what uncertainty components contribute to the constrained forcing range is valid as stated.**

21. p7/193 (section 4.1) It is possible to use an IRF for a specific component from another model, as the authors do, but I am not sure how meaningful this is, since this mixes the climate responses of two different models. To get a consistent model emulation the IRFs for the other RF components should, in principle, be taken from the same model (i.e., NorESM-1).

**We agree that a fully consistent emulation of any given climate model would entail consistent IRFs (and any other parameters) for all species - however there is little evidence**

**that large-scale climate responses are related to aerosol forcing responses in models (e.g., aerosol forcing is not correlated with climate sensitivity in CMIP6 models - Smith et al. 2020 - https://doi.org/10.5194/acp-20-9591-2020) so the experiment we present is reasonable as a sensitivity exercise.**

**Given that two coupled models so far have shown a dramatically different shape for the BC impulse response, our goal is to examine the impact of changing the BC IRF on global temperature. We, therefore, wish to use a realistic BC impulse for this calculation as derived from NorESM instead of postulating something more hypothetical.**

22. p8/239 "it demonstrates nonlinear dynamics" I find this claim unfounded since the nonlinearities in question concern the dynamics of a previously existing model, while the model component contributed by the authors cannot represent the relevant nonlinearity, i.e., that of the climate-carbon cycle feedback.

**Correct, HIRM depends on the parent model–in this case Hector. This limitation has been added to the manuscript in lines 307. We have clarified the wording.**

23. p9/253 (whole paragraph) "most of the linearities" - there is no finding about any specific nonlinearities, and the fact that the Hector SCM has a linear RF-temperature response is no basis for a recommendation for further model development. The nearlinear relationship between RF and temperature is well known and has been demonstrated and exploited in SCMs for a long time (e.g., Joos and Bruno, 1996).

**This paragraph has been modified to reflect this comment.**

24. p9/271 "we demonstrate that the use of a forcing-based impulse response function overcomes most of these limitations." I don't see that any limitations are overcome by this approach.

**Noted, this text has been removed.**

25. p9/273 "These findings imply. . ." Again, there is no basis for such a recommendation.

**Noted, this text has been revised so as to clarify the implications of this work.**

Technical corrections

- p5/132 "In this experiment HIRM was configured" - The word "was" seems to be superfluous here.
**Removed.**

- In Table 1, the unit should be given.
**Added.**

References

- Joos and Bruno, 1996. Pulse response functions are cost-efficient tools to model the link between carbon emissions, atmospheric CO2 and global warming, Phys. Chem. Earth, 21, 471–476.

- Strassmann, K. M. and Joos, F.: The Bern Simple Climate Model (BernSCM) v1.0: an extensible and fully documented open-source re-implementation of the Bern reduced form model for global carbon cycle–climate simulations, Geosci. Model Dev., 11, 1887– 1908, https://doi.org/10.5194/gmd-11-1887-2018, 2018

**Thank you for including these references, very helpful.**

---

## Author Comment (AC3) · 2 Sep 2020

Response to Referees
gmd-2020-33
Reviewer comments in normal type face. **Response in bold.**

Anonymous Referee #3

The authors present a model that couples radiative forcing (potentially from any source, but here solely from the Hector simple climate model) to an impulse response function to calculate global mean surface temperature anomaly, with the possibility to choose different impulse response functions for different forcings (e.g. black carbon).

It is difficult to see what kind of advance this model is. HIRM relies very heavily on Hector, the details of which are documented extensively elsewhere. Using different IRFs for different forcings is not a new concept either (e.g. Richardson et al 2019 for CO2, CH4, BC, SO2 and solar forcing, Larson and Portmann 2016 for volcanic as a special case).

**Indeed, the idea that there is a different climate response depending on the forcing agent has been around for quite some time. For example this has been incorporated mechanistically for aerosols in the MAGICC model for around 30 years now (Wigley. and Raper 1992), and more recently inferred by Shindell (2014) from GCM results. We have expanded our text to add this context. We note that this is a model description paper; we are not claiming that our results are new scientific advances, but rather are presenting this model as a useful tool for rapid analysis and as a testbed for model development and analysis. This is explicitly within scope of journals such as GMD.**

This could be a nice module to include in Hector as an alternative way to calculate temperature in that model. The comparison to the default Hector temperature response function is shown in Figure 2 and seen to be almost identical, so this simplification (is it a simplification?) may be worthwhile. But, due to its nearly total dependence on Hector and the fact that species-dependent efficiacy response functions have been done previously, it doesn't qualify as a brand new model for me. Rather it is a submodule of Hector.

**We have clarified in the revised text that HIRM is independent of Hector, it can be used with input from any model and with any IRF (lines 57-60). We use Hector because it is also open source (so the work here can be replicated by anyone who wishes to do so) and has a convenient interface for obtaining radiative forcing and temperature time series.**

It is possible that there is more to this paper than meets the eye, but if there is it should stand out more, and if the authors believe this does warrant a standalone model, expend some effort in decoupling it from Hector and explain what is improved or new over e.g. Richardson et al. 2019

**We hope the revised text does this. Note, however, that we do not claim that this work is improved over works such as Richardson et al. (2019), which presents an analysis of results from complex models (both coupled and atmosphere-only using a slab ocean). HIRM could certainly be used to quickly examine the implications of the IRFs derived in work such as**

**Richardson et al., but that work is of a different nature than what we present here in a model description paper.**

Specific comments:

Line 18: Examining the effect of aerosol forcing on global temperature: a worthwhile cause. There is not actually done in this paper however. To me this would involve varying the present-day forcing of aerosols, climate sensitivity, and carbon cycle feedbacks, and investigating how this would cause temperature to evolve in a probabilistic fashion. The projections shown in figure 3 are far too constrained as explained in a later comment.

**We certainly agree with the comments of the reviewer in terms of scientific principles. We note, however, that the purpose of this section is to demonstrate how the tool could be used (as part of the GMD model documentation paper), not to conduct a rigorous uncertainty analysis.**

Around line 25, there is a missing link between ESMs and SCMs: Earth System Models of Intermediate Complexity (EMICs). In fact, you could say that in decreasing order of complexity we have ESMs > GCMs > EMICs > SCMs.

**EMICs have been incorporated into the manuscript starting in line 26.**

In the paragraph starting on line 32, the authors discuss the differences between process-based and idealized simple climate models, presumably as a preface for introducing their own model that couples the two components. It is not clear to me that these two concepts are necessarily separate, and if they are, this model may not be as novel as the authors claim. The later versions of FAIR (Smith et al., 2018, GRL) include an impulse response function for CO2 emissions to concentrations and for converting forcing to temperature, and "processed-based" representations of concentrations of greenhouse gases, radiative forcing of GHGs and short-lived climate forcers, and feedbacks from temperature on the carbon cycle and radiative forcing. Leach et al. (2020) in the Generalised Impulse Response model extends the impulse-response framework of the carbon cycle to other greenhouse gases and short-lived climate forcers. In both of these models, the radiative forcing is internally calculated (process-based, in the language of the authors?) and not supplied externally/provided by a different model (as discussed in lines 82-83 for HIRM). For my benefit if not others, could you cite maybe one example of a "process-based" SCM on line 32 if these concepts are separate? MAGICC perhaps?

**We have included MAGICC as an example of a process-based model in line 33. Furthermore the changes made to second and third paragraphs of the introduction section help clarifies the discussion regarding process based and impulse response function based models.**

line 37: "top of troposphere" - this is an old and incomplete definition of radiative forcing, and effective radiative forcing is now preferred - the Smith et al (2018, JGR) reference which is in the bibliography (but not in the text, oddly) goes into some detail on this. I should say this discussion is of limited importance for SCMs.

**We have expanded this discussion slightly to clarify how radiative forcing is defined in Hector (and, therefore, by inference in the examples given in this work).**

line 54: "in addition, the physical interpretation of their behaviour is not always straightforward". Please explain why not.

**This text was confusing and has been removed.**

Line 90: This is a confusing paragraph. On first reading it seems like HIRM doesn't allow for species-dependent efficacy. Then I later read the discussion on BC, and see that the different IRF for BC can be included, which *is* a different efficacy (and time-dependent too). Then on second reading I see that the authors are talking about Hector not having species-dependent efficacy which is more evidence that this model is a component of Hector and not standalone. In general, in section 2.2 it is difficult to follow what the authors did. A flow diagram could help.

**Yes, this section was unclear. HIRM can be configured with a unique IRF for each RF agent. However, for the purposes of the validation exercises HIRM had to be set up the same way as Hector, which only exhibits a single IRF. Text clarifying this point has been added in lines (86, 111, and 307).**

Lines 97-102: It is not correct to exclude carbon cycle feedbacks. It is no good if HIRM can emulate Hector with feedbacks switched off if the latter is not representative of the real world. If the forcing comes from Hector in the first place, feedbacks need not be excluded in the Hector configuration, because HIRM doesn't calculate forcing. The analogy here would be concentration-driven GCM runs, where the concentrations are calculated by MAGICC (which includes carbon cycle feedbacks) but the GCMs themselves do not, going from concentrations > forcing > temperature.

**We've modified the text from lines (123-129) where we discussed the decision to exclude the carbon cycle responses. For the purpose of our validation experiments it is appropriate to exclude them here, but they could be important to include in other applications.**

line 120-121: people have underlying assumptions, but software doesn't

**We respectfully disagree, but this is perhaps just an issue of semantics. Perhaps a better way of expressing it would be that programmers encode their assumptions into the software they create.**

line 124: Hector's IRF - this is the function in figure 1 isn't it, because Hector is not impulse-response based. Could just refer to confirm.

**Correct. Figure 1 is Hector's IRF that was obtained from by running hector as described in the text.**

lines 131-132: it goes without saying that 4xCO2 tests the climate model's response to CO2. The importance of the 4xCO2 experiment is that it can be used to (imperfectly) estimate climate sensitivity, climate feedback and CO2 radiative forcing in GCMs (Gregory et al 2004) by using a forcing with a high enough signal-to-noise ratio to get a clear signal but still small enough to avoid substantial non-linearities and tipping points. Hence it can be used as a line of evidence for climate sensitivity, which itself is an input parameter to many simple climate models. Also, putting the Schwarber reference in line 132 reads like they invented this experiment.

**Correct, it was not our attention to present the Schwarber reference like they invited this experiment, but we see how it could have been interpreted this way. We have modified the text to add more appropriate context.**

line 145: The difference ... and the following sentence, can be dropped. It's apparent from the naked eye that the differences are imperceptible, I don't think this needs to be rigorous. Similar sentence in following paragraph.

**Noted, text in this section has been changed.**

line 157: Which SCM? Hector?

**Noted and corrected.**

lines 161-162: needs a reference

**We now cite Forest 2018.**

line 174: 29000 is a bit of a random number, is there a motivation for this?

**29000 is the number of combinations of the uncertainty scalars when the uncertainty rangers were sampled, this is described in the text starting in line 203.**

lines 176-177: wrong values (-1.9 to -0.1 is AR5 "very likely" i.e. 5-95% estimate), and also wrong citation (Myhre et al., 2013).

**Correct, our range was modified from Myhre et al. 2013, the text in lines 212-213 more clearly describes the range that was used.**

Figure 3: why does uncertainty reduce over time? 2100 temperature is very tightly constrained, but this is the timeframe over which uncertainties in radiative forcing, climate sensitivity and carbon cycle feedbacks multiply. I know these are not sampled, but this should very clearly be stated and the fact that this is not a true future uncertainty quantification of warming. I'd also check the constraints - is 1.6C of warming in 2010, which passes the constraint, realistic?

**The uncertainty decreases by year 2100 because overall aerosol forcing decreases. In the scenario that was used aerosol and precursor emissions decrease substantially over the 20[th]**

century so that, regardless of what is assumed about forcing per unit aerosol/precursor emissions, the overall impact of aerosol forcing is smaller. Therefore, the absolute magnitude of the impact of aerosol forcing uncertainty in 2100 also decreases.

**Thank you for the careful reading of the figure. The upper constrained value of around 1.6C in 2010 is consistent with the applied temperature constraint which is only over 1880–2012. There is a non-trivial amount of positive forcing prior to 1880 due to both well-mixed greenhouse gases and also, potentially, from aerosols (If BC forcing is strong and cooling aerosol forcing is weak). See Smith and Bond 2014, Figure 4. The figure caption has been amended to note this.**

lines 189-190: Important point, long known. Compare/cite some relevant studies e.g. Forest (2018). Figure 4 would be more naturally expressed in terms of a W/m2 aerosol forcing posterior for e.g year 2010, perhaps add a subpanel e. This would back up the claim that strong aerosol forcing values do not pass the constraints.

**Most of these previous studies do not represent uncertainty in the different aerosol components separately. Only Meinshausen et al. 2009 and Tomassini et al 2007 include BC, OC, SO2 direct, and aerosol indirect forcings (Forest 2018, Table S4), but only show them graphically; we have added those references. Note in this example application the aerosol forcing range is supplied as a constraint so its not an independent output to compare to these previous results.**

**Figure 4 and its caption has been updated as per your suggestion.**

Line 214: Mention the perturbation size from line 220 here. Richardson et al. (2019) included a multi-model response for BC and would be more appropriate to use than the single-model study here.

**We agree this would be useful to use the Richardson et al response function. However, the parameters of their multi-model response function for BC was not provided in their paper or SI (and is not yet available from the authors). The perturbation size is mentioned in line 262.**

line 230: maybe a better wording would be "... the global temperature was 0.2C lower using the specific BC IRF from Sand et al. (2015)." Avoid using "decreased" in this sentence.

**Noted and changed.**

minor:

line 9: Earth (rather than earth)
line 29-30: would probably get picked up in proofing but check citation spellings (Meehl, Meinshausen)
line 40: Myhre et al.

line 91: forging
**Noted and changed.**

line 200: also Richardson et al. 2019
**Added.**

line 226: units, should be C/(W/m2)? i.e. the m2 is in the numerator

**It now reads °C W$^{-1}$m$^{-2}$**

line 253: HRIM (and in several other places)
line 244: "a" not required

**Noted and changed.**

line 245: "dynamics" - not really dynamics is it - just a different IRF

**Noted, this text changed to "temperature response".**

line 438: 29000 times

**Noted and changed.**

References:

Richardson et al. 2019: https://agupubs.onlinelibrary.wiley.com/doi/full/10.1029/2019JD030581
Larson and Portmann 2016:
https://journals.ametsoc.org/jcli/article/29/4/1497/35504/ATemporal-Kernel-Method-to-Compute-Effective
Smith et al. 2018, GRL: https://gmd.copernicus.org/articles/11/2273/2018/
Leach et al. 2020: https://gmd.copernicus.org/preprints/gmd-2019-379/
Forster et al. 2016: https://agupubs.onlinelibrary.wiley.com/doi/full/10.1002/2016JD025320
Smith et al. 2018, JGR: https://agupubs.onlinelibrary.wiley.com/doi/full/10.1029/2018GL079826
Gregory et al. 2004: https://agupubs.onlinelibrary.wiley.com/doi/10.1029/2003GL018747
Forest 2018: https://link.springer.com/article/10.1007/s40641-018-0085-2

**Thank you for the suggested references.**

---

## Author Response (AR2)

[revised manuscript text omitted]

Response to Referees
gmd-2020-33
Reviewer comments in normal type face. **Response in bold.**

Ben Kravitz (Referee #1)

**Thank you for taking the time to review this manuscript again and for recommending the manuscript for publication.**

Anonymous Referee #3

Thanks for your considered responses. A few techincal issues I spotted that would possibly get picked up by typesetters:

line 35: Whereas some SCMs are more abstract, consist of fewer highly parameterized equations.

line 120: Hector, BC emissions are converted directly to radiative forcing, and therefore an emissions pulse of BC is analogous to a radiative forcing pulse.

line 149: 3000 [years?]

line 217: suggest that sentence in brackets comes before the period after 0.14 W m-2].(Similar to Myhre et al. 2013, but adjusted to account for nitrate and dust forcing and empirical constraints, see discussion in Smith and Bond 2014).

**Thanks for taking the time to review this manuscript again, the technical corrections you mentioned have been corrected.**